# Novel Roles of RNA m6A Methylation Regulators in the Occurrence of Alzheimer’s Disease and the Subtype Classification

**DOI:** 10.3390/ijms231810766

**Published:** 2022-09-15

**Authors:** Min Li, Wenli Cheng, Luyun Zhang, Cheng Zhou, Xinyue Peng, Susu Yu, Wenjuan Zhang

**Affiliations:** Department of Public Health and Preventive Medicine, School of Medicine, Jinan University, Guangzhou 510632, China

**Keywords:** Alzheimer’s disease, m6A methylation, consensus clustering, biomarker

## Abstract

Alzheimer’s disease (AD) is one of the most common forms of dementia, closely related to epigenetic factors. N6-methyladenosine (m6A) is the most abundant RNA modification, affecting the pathogenesis and development of neurodegenerative diseases. This study was the first exploration of the combined role of 25 common m6A RNA methylation regulators in AD through the integrated bioinformatics approaches. The 14 m6A regulators related to AD were selected by analyzing differences between AD patients and normal controls. Based on the selected m6A regulators, AD patients could be well classified into two m6A models using consensus clustering. The two clusters of patients had different immune profiles, and m6A regulators were associated with the components of immune cells. Additionally, there were 19 key AD genes obtained by screening differential genes through weighted gene co-expression network and least absolute shrinkage and selection operator regression analysis, which were highly associated with important m6A regulators during the occurrence of AD. More interestingly, NOTCH2 and NME1 could be potential targets for m6A regulation of AD. Taken together, these findings indicate that dysregulation of m6A methylation affects the occurrence of AD and is vital for the subtype classification and immune infiltration of AD.

## 1. Introduction

Alzheimer’s disease is the main cause of dementia among the elderly with degeneration of brain cells [1]. Currently, there are about 50 million people diagnosed with AD worldwide, and it is estimated that the number will double every 5 years, reaching 152 million by 2050 [2]. AD is a multifactorial disease associated with several hazard factors, causative and risk genes. Studies of twins showed that the risk of AD was 60~80% dependent on heritable factors [3].

The m6A is the most abundant internal modification on mRNA, dynamically and reversibly regulated by RNA methyltransferases, demethylases and m6A binding proteins, affecting the fates of RNA including its metabolism [4], degradation, translation, splicing and nuclear export, etc. [5]. RNA methyltransferases cause RNA adenylate to undergo methylation modification, and the methyl groups can be removed by RNA demethylases. The methylated reading proteins can recognize the sites of m6A methylation modification and influence RNA function [6].The methyltransferases, also known as “writers”, include methyltransferase-like 3/5/14 (METTL3/5/14) [7,8,9,10], Wilms tumor 1-associated protein (WTAP) [11], RNA-binding motif protein 15/15B (RBM15/15B) [12,13], zinc finger CCCH domain-containing protein 13 (ZC3H13) [14], KIAA1429 [15], Cbl proto-oncogene like 1 (CBLL1) [16], zinc finger CCHC-type containing 4 (ZCCHC4) [17], and phosphorylated CTD interacting factor 1 (PCIF1) [18]. Then, the reverse process is regulated by demethylases, also called “erasers”, including Fat mass and obesity-associated protein (FTO) [19] and Alkb homologue 5 (ALKBH5) [20]. The RNA binding proteins include YT521-B homology (YTH) domain-family protein 1/2 (YTHDC1/2) [21,22] and YTH domain-family protein 1/2/3 (YTHDF1/2/3) [23,24,25]. Besides the YTH domain-containing m6A readers, several additional RNA-binding proteins are preferentially bound to m6A methylated RNA. Among these are eukaryotic translation initiation factor 3 subunit H (EIF3H) [26], fragile X mental retardation 1 (FMR1) [27], insulin like growth factor 2 mRNA binding protein 1/2/3 (IGF2BP1/2/3) [28], heterogeneous nuclear ribonucleoprotein (HNRNPC) [29], and heterogeneous nuclear ribonucleoproteins A2/B1 (HNRNPA2B1) [30].

The m6A is an abundant RNA modification in the brain, participating in the development of nervous system and neurodegenerative diseases [31,32], becoming a new frontier in AD. At present, there are few researches about AD and m6A modulators. Moreover, the role of m6A RNA methylation modulators in AD and the correlative genes have never been reported.

In this study, we first explored and validated 14 m6A methylation regulators associated with AD and identified 19 key AD genes. Further, we comprehensively evaluated the novel role of m6A regulators in AD subtype classification and the correlation between m6A regulator expressions and immune infiltration, and finally explored the potential targets of m6A regulators in AD. The study may provide important evidence for the functions of m6A methylation in AD and offer clues for its subtype classification and immunotherapeutic strategies.

## 2. Results

### 2.1. Expression and Correlation of 25 m6A RNA Methylation Regulators in AD

Differential expression analysis of 25 m6A RNA methylation regulators was performed between AD patients and control samples. As shown in Figure 1A,B, there were 14 vital m6A regulators that had been screened and visualized from the boxplot and heat map. Compared with that in the controls, the expression of m6A regulators increased significantly (*p* < 0.05), including *METTL14*, *RBM15*, *ZCCHC4*, *YTHDC1*, *YTHDF1*, *FMR1*, *IGF2BP2*, and *HNRNPA2B1*, while the expression of some other m6A regulators was significantly decreased, including *METTL3*, *METTL5*, *PCIF1*, *YTHDF2*, *FIF3H*, and *FTO* in AD patients. Overall, the expression level of m6A RNA methyltransferase *METTL14*, *RBM15*, and *ZCCHC4* increased separately, whereas that of *METTL5*, *METTL3*, and *PCIF1* decreased in AD patients. The expression level of m6A RNA demethylase *FTO* decreased. For m6A RNA binding protein, the expression levels of *YTHDC1*, *YTHDF1*, *FMR1*, *IGF2BP2*, and *HNRNPA2B1* rose, while *YTHDF2* and *EIF3H* expression dropped in AD patients, compared with that in controls. Therefore, the level of m6A demethylase *FTO* decreased, while the levels of most of the m6A binding proteins increased in AD patients, indicating that specific m6A modifications might be commonly upregulated in AD.

### 2.2. Two Distinct m6A Patterns Identified by Significant m6A Regulators

Based on the 14 important m6A regulators, the consensus clustering method was utilized to identify different m6A patterns in 167 AD patients. There were two identified m6A patterns, including cluster A and cluster B, as shown in Figure 2A–C. Cluster A had 71 AD patients, and cluster B had 96 AD patients. The differential expression levels of 14 important m6A regulators between the two clusters were then depicted using a boxplot. *METTL5*, *METTL14*, *ZCCHC4*, *YTHDF2*, *EIF3H*, *FMR1*, and *FTO* displayed higher levels in cluster B than in cluster A, while *IGF2BP2* showed the opposite pattern, as shown in Figure 2E. Simultaneously, principal component analysis (PCA) found that the 14 significant m6A regulators could discriminate the two m6A patterns better (Figure 2D).

Then, we analyzed the differences in immune infiltrating cells and Braak staging of AD between the two types of m6A patterns. Cluster A and cluster B had differential immune infiltrating cells with statistical significance (*p* < 0.05). Compared with cluster B, cluster A showed a higher infiltrating proportion in activated B cells, activated dendritic cells, CD56 bright natural killer cells, immature B cells, myeloid-derived suppressor cells (MDSC), mast cells, natural killer cells, neutrophils, T follicular helper cells, type1 T helper cells, and type17 T helper cells, while showing lower proportions of infiltration in activated CD4 T cells and type2 T helper cells, as shown in Figure 3A. The results suggested that cluster A was linked to Th1 and Th17 dominant immunity, while cluster B was associated with Th2 dominant immunity. From Figure 3B, the Braak staging of cluster A was higher than that of cluster B (*p* < 0.01). In addition, single sample gene set enrichment analysis (ssGSEA) was used to calculate the abundance of immune cells in AD samples and evaluate the association of 14 significant m6A regulators with immune cells. We found that three m6A binding proteins, *HNRNPA2B1*, *YTHDF1* and *IGF2BP2*, were positively correlated with many immune cells, while *FTO* was negatively associated with most immune cells, as shown in Figure 3C. Thus, there was a close correlation between the expression levels of the m6A regulators and the immune infiltrating cells in AD patients, especially for *HNRNPA2B1*, *YTHDF1*, *IGF2BP2*, and *FTO*.

### 2.3. Screening and Functional Enrichment Analysis of DEGs

Under the thresholds of |log2 fold change (FC)| > 0.585 and false discovery rate (FDR) < 0.05, a total of 194 differentially expressed genes (DEGs) were obtained from the training dataset, including 124 down-regulated and 70 up-regulated genes, as shown in Figure 4A. Furthermore, the expression levels of all of the DEGs were displayed in a heatmap (Figure 4B), and these genes were well clustered between AD patients and normal controls.

To gain deeper insight into the biological functions of DEGs, Gene Ontology (GO) annotation and Kyoto Encyclopedia of Genes and Genomes (KEGG) pathway enrichment analyses were performed. The top 10 enriched GO terms are shown in Figure 4C and were categorized into three parts: biological process (BP), cellular component (CC) and molecular function (MF) [33]. DEGs of BP were involved in cellular divalent inorganic cation homeostasis, renal system development, urogenital system development, kidney development, nephron development, nephron tubule development, and renal tubule development. CC analysis revealed that DEGs were markedly enriched in the distal axon, glutamatergic synapse, axon terminus, neuron projection terminus, mitochondrial outer membrane, synaptic vesicle, and synaptic vesicle membrane. For MF, the top two significantly enriched terms were guanosine triphosphate (GTP) binding and guanyl nucleotide binding. The enriched KEGG pathways are presented in Figure 4D, including AD, cyclic adenosine monophosphate (cAMP) signaling pathway, neuroactive ligand–receptor interaction, Parkinson’s disease, phagosome, dopaminergic synapse, mineral absorption, and gap junction.

### 2.4. Identification of Key AD Genes

In order to confirm the modules and genes associated with the occurrence of AD, we performed weighted gene co-expression network analysis (WGCNA) of genes obtained from the merged dataset according to the thresholds of FDR < 0.05. According to the fitting index of the scale-free network and the mean connectivity, *β* = 3 was calculated and selected as the soft threshold of this dataset, and the adjacency matrix and TOM matrix were calculated. Subsequently, nine modules were generated, including green, magenta, purple, pink, black, turquoise, yellow, blue, and brown, as shown in Figure 5A. To identify the modules correlated with AD status, we analyzed the relationship between each module and AD. As shown in Figure 5B,C, from the correlation and importance between the nine modules and AD, the turquoise module (*r* = 0.37, *p* < 0.001), the black module (*r* = 0.29, *p* < 0.001), and the magenta module (*r* = −0.29, *p* < 0.001) ranked as the top three, which were considered to be closely related to the occurrence of AD. These three important gene modules were further used as key modules for gene significance (GS) and module membership (MM) analysis. Based on the criteria of GS > 0.2 and MM > 0.8, the 129 hub AD genes were screened from turquoise, green, black, and magenta modules (Figure 5D–F).

The 42 intersecting genes were acquired by taking the intersection of 194 DEGs and 129 hub AD genes (Figure 6A). To explore the biomarkers of AD more accurately, we performed the feature screening of 42 intersecting genes through the least absolute shrinkage and selection operator (LASSO) regression. According to the optimum *λ* value, the 19 genes were identified as key genes of AD, including NME/NM23 nucleoside diphosphate kinase 1 (*NME1*), phospholipid scramblase 4 (*PLSCR4*), ATPase H+ transporting V1 subunit G2 (*ATP6V1G2*), regulatory factor X4 (*RFX4*), glucose-6-phosphate isomerase (*GPI*), mal T cell differentiation protein 2 (*MAL2*), notch receptor 2 (*NOTCH2*), C1q and TNF related 4 (*C1QTNF4*), yes-associated protein 1 (*YAP1*), calcyon neuron specific vesicular protein (*CALY*), sulfotransferase family 4A member 1 (*SULT4A1*), G protein subunit gamma 3 (*GNG3*), SRY-Box transcription factor 9 (*SOX9*), sodium voltage-gated channel beta subunit 3 (*SCN3B*), PNMA family member 2 (*PNMAL2*), Cholecystokinin (*CCK*), Actin like 6B (*ACTL6B*), malate dehydrogenase 1 (*MDH1*), and synuclein beta (*SNCB*), as shown in Figure 6B,C.

### 2.5. Co-Expression Relationship of the 14 Significant m6A Regulators and the 19 Key AD Genes

To further investigate the roles of the 14 significant m6A methylation regulators and the 19 key genes in AD, we analyzed their co-expression relationships. The Protein–protein interaction (PPI) network showed that these significant m6A methylation regulators could interact with some of the key AD genes directly or indirectly, as shown in Figure 7A. In addition, the key AD genes, *NOTCH2* and *NME1*, were closely related to m6A regulators and might be the important targets in m6A methylation during the progression of AD.

### 2.6. Roles of NOTCH2 and NME1 in AD and Their Association with m6A Regulators

In order to verify whether NOTCH2 and NME1 were related to AD, we evaluated their expressions in healthy controls and AD patients. As shown in Figure 7B,C, the mRNA level of *NOTCH2* was higher (*p* < 0.001), whereas the level of *NME1* was lower (*p <* 0.001), in AD patients, compared with that in the normal samples. Additionally, the expression level of *NOTCH2* was negatively correlated with *YTHDF2* (*r* = −0.412, *p* < 0.001), *FTO* (*r* = 0.341, *p* < 0.001), *PCIF1* (*r* = 0.324, *p* < 0.001), *METTL5* (*r* = 0.191, *p* < 0.001) and *ZCCHC4* (*r* = 0.119, *p* < 0.001) with significance, while positively correlated with *HNRNPA2B1* (*r* = 0.423, *p* < 0.001), *IGF2BP2* (*r* = 0.409, *p* < 0.001), and *YTHDC1* (*r* = 0.352, *p* < 0.01), as shown in Figure 7D. Interestingly, the levels of *NME1* were also associated with these m6A regulators, while *NOTCH2* had completely opposite results (Figure 7E). The above results suggested that the altered expressions of *NOTCH2* and *NME1* in AD brain tissues were tightly related to the m6A regulators.

To verify the expression profile of NOTCH2 and NME1 in AD and their relationships with m6A regulators, we adopted an external validation dataset (GSE122063) with 100 samples in all to further verify the results. Similarly, there was a significant increase in *NOTCH2* (*p* < 0.05) and an obvious decrease in *NME1* in AD patients (*p* < 0.001), as shown in Figure 8A,B. From Figure 8C,D, the expression levels of *NOTCH2* and *NME1* in the AD brain tissues were regulated by m6A regulators, which was consistent with the above results.

## 3. Discussion

Abnormal m6A modifications may lead to disorders in important genes that regulate key cellular processes and disrupt homeostasis related to disease. The m6A RNA methylation modulators contribute to the development of many cancers, such as lung, liver, breast cancer, etc. [34,35,36,37], and also affect the prognosis of various cancers [38,39]. However, few studies focus on the relationship between m6A RNA methylation and the brain. Several studies have shown that m6A regulators were associated with AD. The m6A regulators exerted a critical function in both early and late brain development in a spatiotemporal fashion, and controlled the protein levels of key genes involved in AD-associated pathways [32]. However, the mechanisms of m6A regulators in human AD have not been fully explored. This study sought to integrate bioinformatics tools to investigate the involvement of the 25 prevalent m6A RNA methylation regulators in AD for the first time. 

We identified 14 important m6A regulators based on differential expression analysis between normal and AD brain tissues. Compared with those in normal controls, the expression levels of *METTL14*, *RBM15*, *ZCCHC4*, *YTHDC1*, *YTHDF1*, *FMR1*, *IGF2BP2*, and *HNRNPA2B1* were higher in AD patients, while *METTL3*, *METTL5*, *PCIF1*, *YTHDF2*, *FIF3H*, and *FTO* were lower correspondingly. Most of the m6A methylation binding proteins, including *YTHDC1*, *YTHDF1*, *FMR1*, *IGF2BP2* and *HNRNPA2B1*, were elevated in AD brain tissues, suggesting that m6A modifications of specific target genes were generally increased in AD brain tissues. Differential levels of these m6A regulators would bring the abnormal epi-transcriptome microenvironment, also affect the expression of target genes, and ultimately contribute to the onset and progression of AD. The m6A modifications played key roles in the synaptic function of brain neurons, as the main mechanisms of AD development [40,41]. However, m6A and the intrinsic regulatory mechanisms remain blank in AD. Our results have provided the theoretical basis and possible research direction of m6A to the etiology, clinical application, and potential molecular therapy for AD. For normal memory and learning functions, METTL3 and METTL14 were fundamentally related to long-term memory formation and normal striatal learning functions in human and mouse hippocampus [32]. FTO has been shown to promote insulin-deficiency-associated AD by decreasing TSC complex subunit 1 (TSC1) mRNA levels, activating the mammalian target of the rapamycin (mTOR) signaling pathway, and promoting tau protein phosphorylation [31]. The relationship between other m6A regulators and AD is still limited to changes in macroscopic levels and lacking minimally specific mechanistic studies.

Based on the 14 important m6A regulators, m6A patterns were identified in cluster A and cluster B, which were able to distinguish AD subtype well. Furthermore, immune infiltration was also an important factor in the development of AD, affecting the inflammatory response as a key role in its pathogenesis. The differences in immune infiltrating cells of the two m6A patterns showed that the levels of activated B cells, dendritic cells, CD56 bright natural killer cells, immature B cells, MDSC, mast cells, natural killer cells, neutrophils, T follicular helper cells, type 1 T helper cells, and type 17 T helper cells were significantly higher in cluster A than those in cluster B, while the activated CD4 T cells and type 2 T helper cells were lower. Cluster A was associated with Th1 and Th17 dominant immunity, while cluster B was associated with Th2 dominant immunity. After comparing the two types of patients, we found that the Braak staging of cluster A was higher than that of cluster B. Neuroinflammation is a key regulator of AD pathogenesis, and abnormal T cells contribute to it by direct crosstalk with glial cells infiltrating the brain and secreting pro-inflammatory mediators. Th1 and Th17 cells are two major pro-inflammatory T cell subtypes elevated in AD generally. In contrast, Th2 cells reduce the inflammatory function of microglia and promote a neuro-supportive microenvironment [42]. Therefore, the severity of cluster A is higher than that of cluster B, and m6A regulators can be well classified into two types of AD. In addition, ssGSEA was used to calculate the abundance of immune cells in AD and assess the association of 14 significant m6A regulators with immune cells. Three m6A methyltransferases, HNRNPA2B1, YTHDF1 and IGF2BP2, were positively associated with many immune cells, whereas FTO was negatively associated with most immune cells. Therefore, m6A regulators are associated with inflammation and the immune microenvironment of AD, contributing to its occurrence.

Moreover, a total of 194 DEGs were obtained from the training dataset, including 124 down-regulated and 70 up-regulated genes. GO analysis showed that the DEGs were enriched in cellular divalent inorganic cation homeostasis, renal system development, etc., which suggested that the expression of these abnormal genes might influence the development of AD via the aforementioned biological processes. This further confirmed the association between brain activity and kidney function. Clinical studies had demonstrated that patients with chronic kidney disease (CKD) were more prone to develop cognitive impairment and AD, and that the degree of cognitive impairment was closely associated with the progression of CKD and renal failure [43]. DEGs were shown to be involved in AD, the cAMP signaling pathway, neuroactive ligand–receptor interaction, dopaminergic synapse, and other pathways. However, cAMP signaling was the most enriched pathway and a crucial second messenger regulating a variety of intracellular processes and neurobehavioral activities [44], and reduction of its level was associated with the pathogenesis of various neurodegenerative diseases including AD [45]. We then analyzed and identified 129 AD hub genes using the WGCNA method and combined 194 differential genes to obtain 42 intersecting genes, which were differentially expressed AD hub genes. Further, the 19 key AD genes were screened from these 42 genes based on the best *λ* value using LASSO-Cox regression analysis.

Then, the co-expression relationships of 14 m6A regulators with 19 AD key genes were analyzed to investigate their associations. From the PPI network, these significant m6A regulators interacted with some key AD genes directly or indirectly. More importantly, we found that NOTCH2 and NME1 might be the key links between m6A methylation and the regulation of AD. The expression levels of *NOTCH2* were upregulated, and *NME1* was downregulated in the AD brain tissues, closely related to m6A regulators. To eliminate the bias caused by the single training dataset, we added another dataset for validation, which was consistent with the results derived from the training dataset.

The NOTCH signaling pathway is one of the most important pathways, determining cell fate with roles in a variety of developmental processes, including neurogenesis, somatogenesis, and others. Aberrant NOTCH signaling contributes to a growing number of pathological processes, including liver, heart, and neurological diseases, and multiple cancer types. NOTCH2 is a major regulator of neural stem cell activity in the adult brain, required for cell quiescence throughout aging, and conditional knockout of NOTCH2 can rejuvenate neurogenesis in the hippocampal dentate gyrus of aged mice [46]. In AD brains, NOTCH1 interacts with presenilin-1 and amyloid precursor protein (APP), by a 2-fold increase in expression in the hippocampus compared to that in normal control. NOTCH2 also interacts with APP [47,48]. However, their relationship with AD pathology remained unclarified. We suspect that in early and late onset AD [49], abnormal cell cycle re-entry of neurons may be an upstream event leading to NOTCH signaling activation, involved in neurodegeneration. NME1 is a protein with serine/threonine-specific protein kinase activity in neural development. Overexpressed NME1 has induced neuronal differentiation in glial progenitor cells, while its downregulation leads to an increase in the number of oligodendrocytes [50]. More interestingly, the proneural effects of NME1 are mediated by the cAMP-dependent protein kinase signaling pathway, closely related to AD [51]. Therefore, m6A may have effects on the occurrence of AD and modify NOTCH2 and NME1. The specific roles of these RNA methylation regulators and the potential target genes, as well as the way they affect these genes and their aberrant interactions in the development of AD, need to be explored further.

Therefore, m6A regulators play a vital role in the occurrence, subtype classification and immune infiltration of AD. The key AD genes, also including *NOTCH2* and *NME1*, may be potential m6A methylation targets, providing new clues for the prevention and intervention of AD.

## 4. Materials and Methods

### 4.1. Data Acquisition and Preprocessing

Gene expression data from the datasets GSE48350, GSE5281 and GSE122063 were downloaded from the Gene Expression Omnibus [52]. The data in GSE48350 and GSE5281 were obtained using the GPL570 platform, and the samples were all brain tissue sections. These two datasets from the same platform were merged into a new dataset as a training group, including 250 normal controls and 167 AD patients. Batch rectification of the two datasets was performed using the “sva” and “limma” package (version 3.50.1). The external validation data, GSE122063, were obtained using the GPL16699 platform, including 56 AD patients and 44 normal brain tissue samples. Gene expression profiles were normalized using the “normalize Between Arrays” function of the “limma” package (version 3.50.1) in R. Additionally, Braak staging data of AD were extracted from GSE5281 and used for further analysis.

### 4.2. Selection of m6A RNA Methylation Regulators

We extracted 25 common m6A regulatory factors from the genome-wide expression dataset, as shown in Table 1, and identified significant m6A regulatory factors by analyzing the differences between non-AD and AD patients in the training dataset. Differential expression of 25 m6A methylation regulators between AD patients and normal controls was analyzed by the “limma” package (version 3.50.1). The *p* < 0.05 was considered statistically significant.

### 4.3. Identification of Molecular Subtypes Based on the Significant m6A Regulators

Consensus clustering is a resampling-based approach to identifying each member and their subgroup number, as well as verifying the rationality of clustering. Consensus clustering was used to discriminate different m6A patterns based on essential m6A regulators using the R package “ConsensusClusterPlus” (version 1.60.0) [53].

### 4.4. Estimation of Immune Cell Infiltration

ssGSEA was performed using the “GSVA” package (version 1.44.2) to assess immune cell abundance in AD patients. The gene expression levels in the samples were first sequenced using ssGSEA. Next, these genes were searched in the input dataset, and then the expression levels were summarized. Based on the above evaluation, the abundance of immune cells was obtained in each sample.

### 4.5. Screening and Functional Enrichment Analysis of DEGs

The “limma” package (version 3.50.1) in R software was used to identify DEGs between AD patients and healthy controls. The *p* < 0.05 was applied as the cut-off for DEG screening after being corrected by FDR and |log2 fold change (FC)| > 0.585. The statistically significant DEGs were then used to undertake GO and KEGG pathway enrichment analysis in R software using the “clusterProfiler” package (version 4.4.4). Adjusted *p* < 0.05 was regarded as the cut-off criterion.

### 4.6. Identification of Co-Expression Modules and Key AD Genes

FDR < 0.05 was used as the cut-off value for screening differential genes and “WGCNA” package (version 1.69) was used to identify the obtained mRNA expression data of differential genes [54]. First, samples were grouped to find whether any clear outliers were present. Second, the co-expression network was built using the automatic network construction. The soft thresholding power was calculated using the R function “pickSoftThreshold”, and the co-expression similarity was raised to compute adjacency. Third, to detect modules, hierarchical clustering and the dynamic tree cut function were utilized. Fourth, to link modules to clinical features, GS and MM were determined. For further study, the matching module gene information was collected. In clinically relevant gene module networks, we classified hub genes as genes with a GS > 0.2 and the MM > 0.8. DEGs were intersected with hub AD genes to obtain the intersected genes. Finally, the key AD genes were screened from the intersection genes by LASSO regression analysis using “glmnet” package (version 4.1.4) in R software.

### 4.7. PPI Network Construction

PPI networks were constructed by a search tool for the retrieval of interacting genes (STRING) online (https://cn.string-db.org/cgi/input.pl, accessed on 2 August 2022) to analyze the interaction between the significant m6A regulators and key AD genes. Confidence >0.25 was used as the filter condition. The PPI network hid disconnected proteins and was visualized by STRING.

### 4.8. Statistical Analysis

The *t*-test was used to compare the expression levels of m6A regulators between two different groups. The Wilcoxon rank sum test was applied to the Braak stage differences. The correlations within the continuous variables were analyzed by the Pearson’s test. All calculations in the study were dependent on R version 4.2.1(R Core Team. Auckland, New Zealand) and Microsoft Excel 2016 (Microsoft corp., Washington, DC, USA). The statistical significance was defined as a two-sided *p* < 0.05.

## 5. Conclusions

In conclusion, the m6A methylation regulators have played crucial roles in the occurrence, subtype classification, and immune infiltration of AD. There were 14 m6A regulators associated with the development of AD, and then AD patients could be well classified into two categories, one of which was closely associated with Th1 dominated immunity with a higher Braak staging. Furthermore, HNRNPA2B1, YTHDF1, IGF2BP2A, and FTO were closely associated with AD immune cell components. The m6A regulators and key AD genes have close connections, with NOTCH2 and NME1 as the potentially specific modification targets, providing new clues for the precise prevention and intervention of AD. It is imperative to explore more detailed mechanisms of m6A methylation in AD occurrence and immunity changes in future researches.

## Figures and Tables

**Figure 1 ijms-23-10766-f001:**
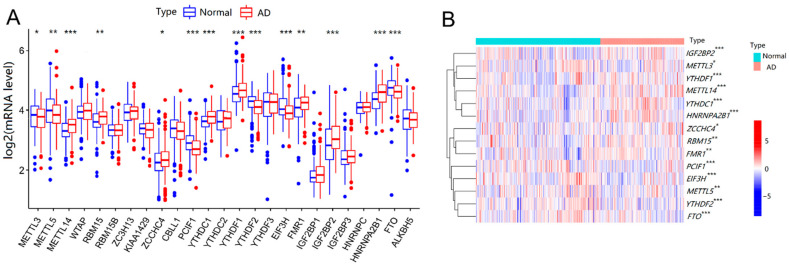
Expressions of m6A methylation regulators between normal controls and AD patients. (**A**) Differential expression boxplot of the 25 identified m6A regulators. (**B**) Expression heat map of 14 differential m6A regulators. The *t*-test was used to analyze the difference in the expression levels of m6A regulators. Compared with the controls, * *p* < 0.05, ** *p* < 0.01, and *** *p* < 0.001.

**Figure 2 ijms-23-10766-f002:**
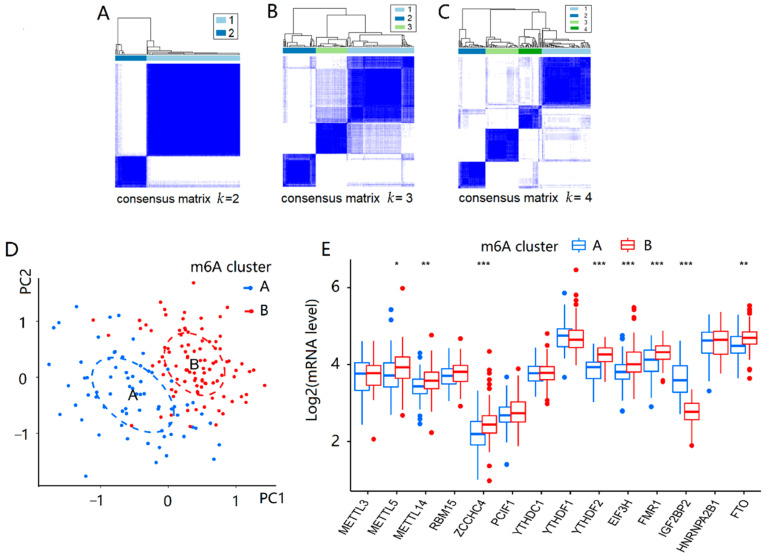
Consensus clustering of 14 important m6A modulators in AD patients. (**A**–**C**) Consensus matrices of the 14 significant m6A regulators for *k* = 2–4. (**D**) PCA for the expression profiles of the 14 significant m6A regulators, with remarkable differences in transcriptomes between the two m6A patterns. (**E**) Differential expression boxplot of the 14 significant m6A regulators in cluster A and cluster B. The *t*-test was used to analyze the differences in the expression levels of the 14 significant m6A regulators between cluster A and cluster B. The results with significant differences were marked as * *p* < 0.05, ** *p* < 0.01, and *** *p* < 0.001.

**Figure 3 ijms-23-10766-f003:**
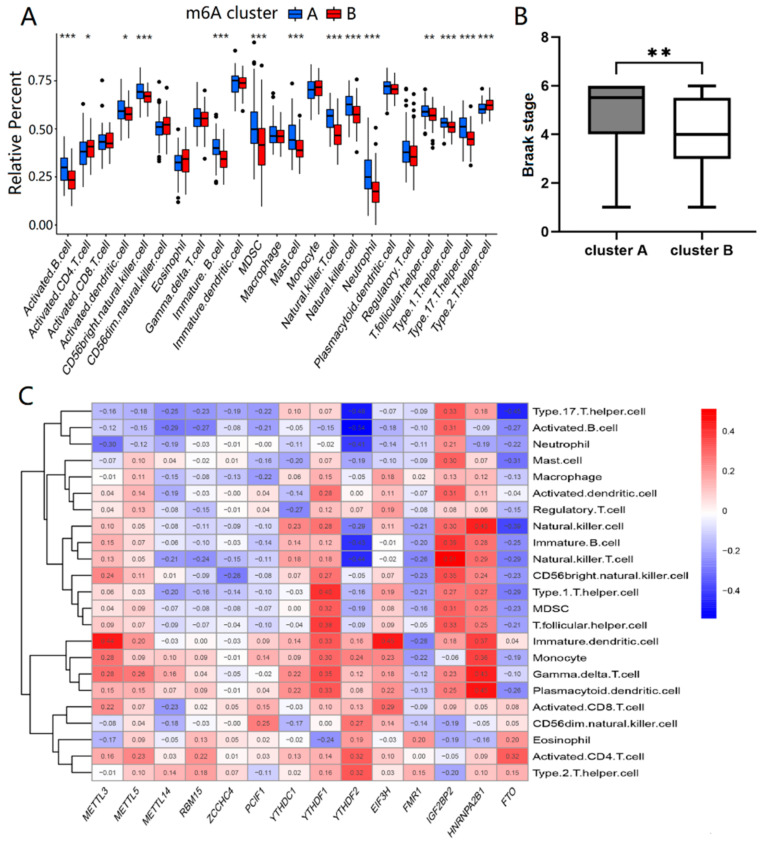
Immune infiltration and Braak staging of two different m6A patterns. (**A**) The *t*-test was performed to analyze the differential immune infiltrating cells between cluster A and cluster B. (**B**) The Wilcoxon rank sum test was performed to compare the Braak stage differences between cluster A and cluster B. (**C**) The correlation between infiltrating immune cells and the 14 significant m6A regulators was assessed by Pearson’s test. The results with significant difference were marked as * *p* < 0.05, ** *p* < 0.01, and *** *p* < 0.001.

**Figure 4 ijms-23-10766-f004:**
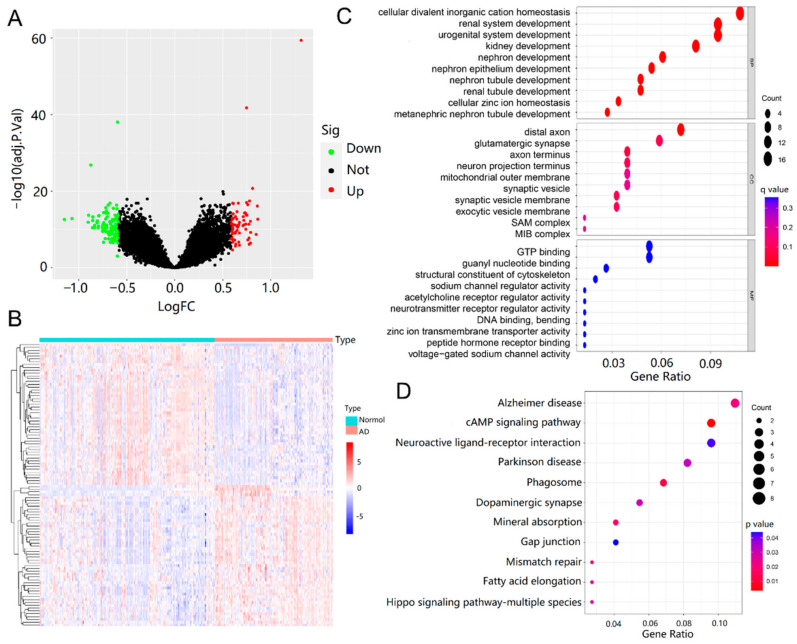
Screening and functional enrichment analysis of DEGs. (**A**) Volcano plot of DEGs (|log2FC| > 0.585 and FDR < 0.05). Up-regulated genes are colored in red and down-regulated genes are colored in green. (**B**) Heatmap of top 70 up and top 124 down DEGs of AD. (**C**) GO enrichment analysis of DEGs (*p* < 0.05 and *q* < 0.05). (**D**) KEGG pathway enrichment results for DEGs (*p* < 0.05 and *q* < 1).

**Figure 5 ijms-23-10766-f005:**
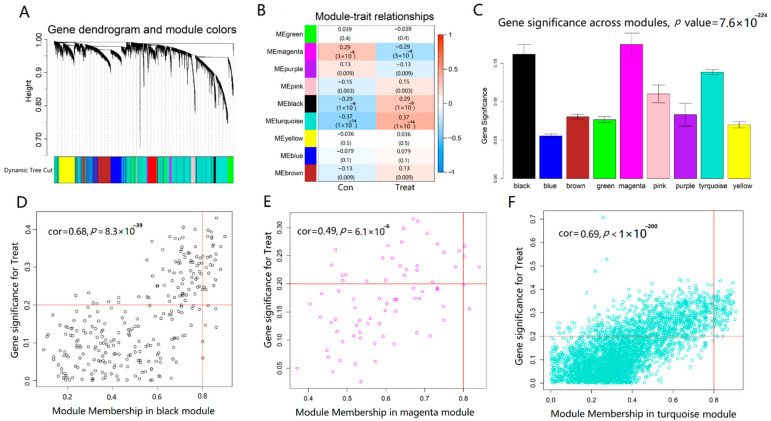
Identification of modules associated with AD through WGCNA. (**A**) Dendrogram of all differentially expressed genes clustered based on a dissimilarity measure. (**B**) Heatmap of the correlation between module eigengenes and AD. Each cell contained the correlation coefficient and *p* value. (**C**) Distribution of average gene significance and errors in the modules associated with AD. (**D**–**F**) Module trait relationship and module features of GS and MM. Every point defined a specific gene within every module that was plotted on the y-axis and the x-axis by GS and MM, respectively.

**Figure 6 ijms-23-10766-f006:**
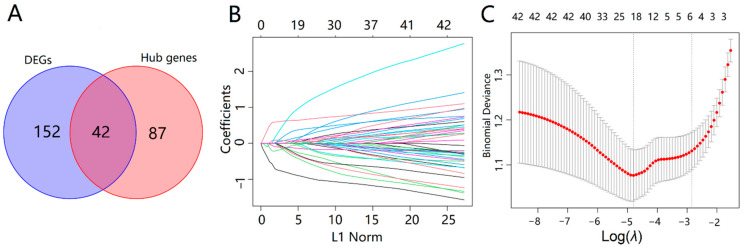
Screening of key genes of AD. (**A**) Venn diagram showed 42 overlapping genes in DEGs and hub genes; (**B**) LASSO coefficient profiles of 42 overlapping genes. (**C**) A coefficient profile plot was produced against the log (*λ*) sequence in the LASSO model. The optimal parameter (λ) was selected as the first black dotted line indicated.

**Figure 7 ijms-23-10766-f007:**
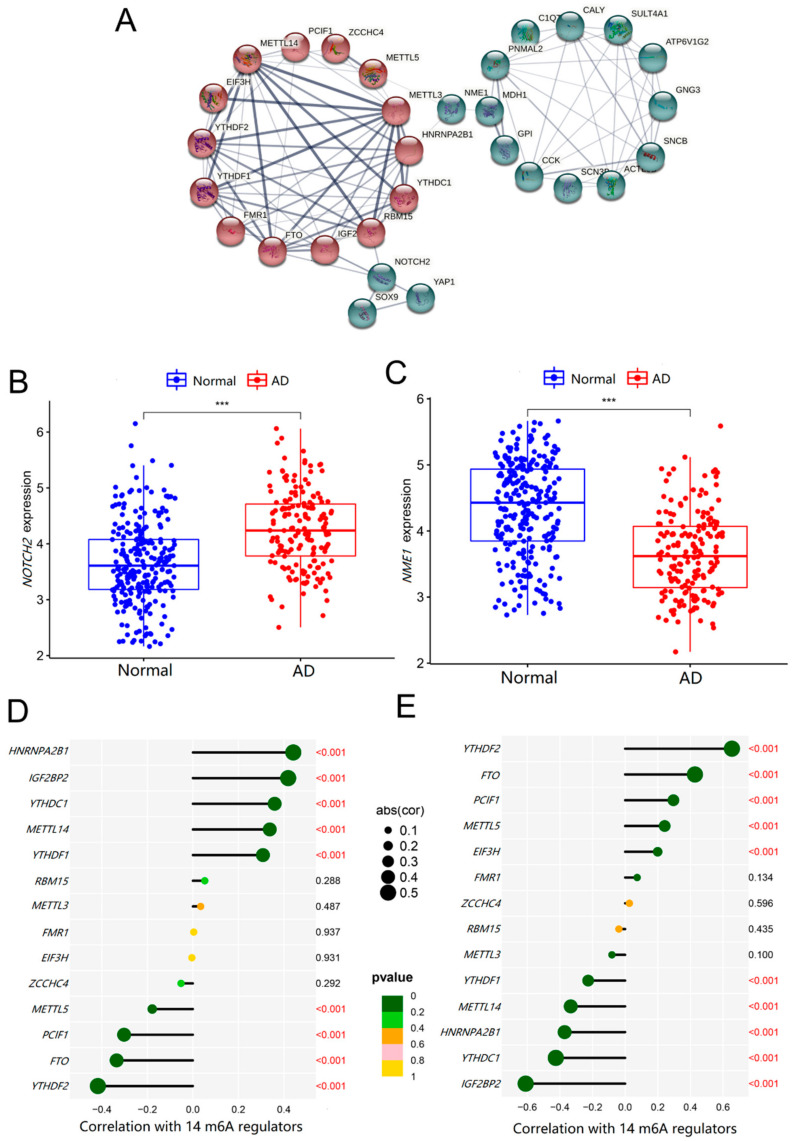
The relationships of m6A regulators and key AD genes. (**A**) PPI network was constructed by 14 m6A regulators as well as 19 key AD genes, with the disconnected proteins being hidden. Wider lines indicate stronger evidence of protein interactions. (**B**,**C**) The *t*-test was conducted to analyze the difference in the expression of *NOTCH2* and *NME1* between normal controls and AD patients. (**D**,**E**) The Pearson’s test was performed to analyze the correlation between *NOTCH2* and *NME1* and m6A regulators. The results with significant differences was marked as *** *p* < 0.001.

**Figure 8 ijms-23-10766-f008:**
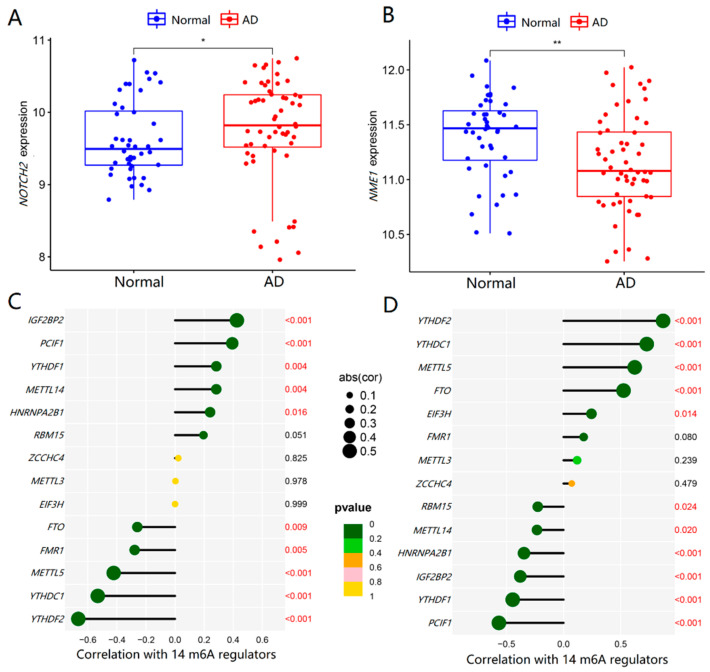
Validation of *NOTCH2* and *NME1* expression in AD and their relationships with m6A regulators. (**A**,**B**) The *t*-test was conducted to analyze the differences in the expression of *NOTCH2* and *NME1* between normal controls and AD patients by using the validation dataset GSE122063. (**C**,**D**) The Pearson’s test was performed to analyze the correlations between *NOTCH2* and *NME1* and m6A regulators. The results with significant differences were marked as * *p* < 0.05 and ** *p* < 0.01.

**Table 1 ijms-23-10766-t001:** List of the 25 m6A regulators included in the current study.

Categories	m6A Regulators	Mechanisms	References
m6A writers	METTL3	Catalyzes m6A modification	[7]
	METTL5	Affects the selection of methylation sites	[10]
	METTL14	Assists METTL3 in recognizing the substrates substraction	[8]
	WTAP	Promotes METTL3-METTL14 heterodimer to the nuclear speckle	[11]
	RBM15/15B	Binds the m6A complex and recruits it to special RNA site	[12,13]
	ZC3H13	Bridges WTAP to the mRNA-binding factor Nito	[14]
	KIAA1429	Guides methyltransferase components to specific RNA regions	[15]
	ZCCHC4	Adds m6A to18S and 28S ribosomal RNAs	[17]
	CBLL1	Binds the m6A evolutionary conserved protein complex	[16]
	PCIF1	Catalyzes m6A methylation on 2-O-methylated adenine	[18]
m6A erasers	FTO	Removes m6A modification	[19]
	ALKBH5	Removes m6A modification	[20]
m6A readers	YTHDC1	Promotes RNA splicing and translocation	[21]
	YTHDC2	Enhances the translation of target RNA	[22]
	YTHDF1	Promotes mRNA translation	[23]
	YTHDF2	Reduces mRNA stability	[25]
	YTHDF3	Mediates the translation or degradation	[24]
	EIF3H	Interacts with METTL3 and enhances translation	[26]
	FMR1	Affects the nuclear export of m6A-modified RNA targets	[27]
	IGF2BP1/2/3	Enhances mRNA stability	[28]
	HNRNPC	Mediates mRNA splicing	[29]
	HNRNPA2B1	Promotes primary microRNA processing	[30]

## Data Availability

The datasets generated and/or analyzed during the current study are openly available in the Gene Expression Omnibus (GSE48350, GSE5281, GSE122063) at https://www.ncbi.nlm.nih.gov/geo/, accessed on 20 July 2022.

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
