# Peer review of "Novel Roles of RNA m6A Methylation Regulators in the Occurrence of Alzheimer’s Disease and the Subtype Classification"

_ijms, 2022, doi:10.3390/ijms231810766_

Round 1

Reviewer 1 Report

Li et al., studied relations between RNA m6A methylation and Alzheimer’s disease using bioinformatics tools.

They provide new information on potential relationship between RNA methylation, differentially expressed genes, inflammation and Alzheimer disease.

The study is important and the results can provide starting point for further more focused studies.

The manuscript however contains lot of formal issues that need to be addressed before publishing.

First, there are many formulations that are not correct in English language and several sentences need to be reformulated to make the text understandable. I would strongly suggest revising entire text for correct English formulations.

Focus on sentences in lines: 93-95, 108-116, 122-123, 164-167, 199-201, 250-254, 305-307.

Further issues to be addressed:

Line 24 – what does ‘aberrant’ mean? It is not clear in this sentence.

Lines 33-34 – why do you use ‘was’ and not ‘is’?

Figure 1 – reformulate figure legend. ‘A’ is the histogram, ‘B’ is heat map, in ‘B, you compare only 14 genes, not 25.

Line 95 – what does ‘cases’ mean? Cases of what?

Line 153 – not ‘blue’, but ‘green’.

From line 178 and figure 6, the text contains wrong references to the figures, e.g. in line 179 there is ‘Figure 7A’, but it should be ’6A’. The numbering in text does not correspond with numbering of figures 6, 7, 8.

Line 180 – What does it mean ‘key’? How were these 19 genes chosen?

Line 270 – What does word ‘two’ refer to? ‘Two’ what?

Line 368 – Table 1, not Table 2.

In general, in figure legends are missing informations about what statistical test was/were used and information on significances.

In general, when you introduce acronym for the first time you need to include full wording of the term. For example in line 168, GS and MM.

Author Response

Reviewer 1:

Li et al., studied relations between RNA m6A methylation and Alzheimer’s disease using bioinformatics tools. They provide new information on potential relationship between RNA methylation, differentially expressed genes, inflammation and Alzheimer disease. The study is important and the results can provide starting point for further more focused studies. The manuscript however contains lot of formal issues that need to be addressed before publishing.

  1. First, there are many formulations that are not correct in English language and several sentences need to be reformulated to make the text understandable. I would strongly suggest revising entire text for correct English formulations. Focus on sentences in lines: 93-95, 108-116, 122-123, 164-167, 199-201, 250-254, 305-307.

→ We thank the reviewer very much for the good suggestion and strict request. Accordingly, we have carefully checked and revised the entire text for correct English presentation. We have also rephrased the following sentences in the new revised manuscript:93-95、108-116、122-123、164-167、199-201、250-254、305-307. Please check them in the new revision. Now we also show them here as below for your convenience and the relative changes are marked in red color.

From line 93 to 95 → From line 91 to 94

Based on the 14 important m6A regulators, the consensus clustering method was utilized to identify different m6A patterns in 167 AD patients. There were two identified m6A patterns, including cluster A and cluster B, as shown in Figure 2A-C. The cluster A had 71 AD patients and the cluster B had 96 AD patients separately.

From line 108 to 116 → From line 110 to 119

Then, we analyzed the differences in immune infiltrating cells and Braak staging of AD between the two kinds of m6A patterns. The cluster A and cluster B had differential immune infiltrating cells with statistical significance (p<0.05). Compared with cluster B, cluster A showed a higher infiltrating proportion in activated B cell, activated dendritic cell, CD56bright natural killer cell, immature B cell, myeloid-derived sup-pressor cells (MDSC), mast cell, natural killer cell, neutrophil, T follicular helper cell, type1 T helper cell, and type17 T helper cell respectively, while lower in activated CD4 T cell and type2 T helper cell, as shown in Figure 3A. The results suggested that cluster A was linked to Th1 and Th17 dominant immunity, while cluster B was associated with Th2 dominant immunity.

From 122 to 123 → From line 125 to 127

Thus, there was a close correlation between the expression levels of the m6A regulators and the immune infiltrating cells in AD patients, especially for HNRNPA2B1, YTHDF1, IGF2BP2, and FTO.

From line 164 to 167 → From line 172 to 175

As shown in Figure 5B and 5C, from the correlation and importance between the 9 modules and AD, the turquoise module (r=0.37, p<0.001), the black module (r=0.29, p<0.001), and the magenta module (r=-0.29, p<0.001) ranked the top three, which were considered to be closely related to the occurrence of AD.

From line 199 to 201 → From line 209 to 211

In addition, the key AD genes, NOTCH2 and NME1 were closely related to m6A regulators, which might be the important targets in m6A methylation during the progression of AD.

From line 250 to 254 → From line 260 to 263

Compared with those in normal controls, the expression levels of METTL14, RBM15, ZCCHC4, YTHDC1, YTHDF1, FMR1, IGF2BP2, and HNRNPA2B1 were higher respectively in AD patients, while METTL3, METTL5, PCIF1, YTHDF2, FIF3H, and FTO were lower correspondingly.

From line 305 to 307 → From line 317 to 321

We then analyzed and identified 129 AD hub genes using WGCNA method and combined 194 differential genes to obtain 42 intersecting genes, which were differentially expressed AD hub genes. Further, the 19 key AD genes were screened from these 42 genes based on the best λ value using LASSO-Cox regression analysis.

  1. Further issues to be addressed:

Line 24 – what does ‘aberrant’ mean? It is not clear in this sentence.

Lines 33-34 – why do you use ‘was’ and not ‘is’?

Figure 1 – reformulate figure legend. ‘A’ is the histogram, ‘B’ is heat map, in ‘B, you compare only 14 genes, not 25.

Line 95 – what does ‘cases’ mean? Cases of what?

Line 153 – not ‘blue’, but ‘green’.

From line 178 and figure 6, the text contains wrong references to the figures, e.g. in line 179 there is ‘Figure 7A’, but it should be ‘6A’. The numbering in text does not correspond with numbering of figures 6, 7, 8.

Line 180 – What does it mean ‘key’? How were these 19 genes chosen?

Line 270 – What does word ‘two’ refer to? ‘Two’ what?

Line 368 – Table 1, not Table 2.

→ We appreciate the reviewer’s good suggestions and strict requests. According to the requirements and instructions, we have already checked the original text carefully and corrected all of these issues in the new revised manuscript. We also show them here in red color as follows for your check conveniently.

About Line 24 – what does ‘aberrant’ mean? It is not clear in this sentence.

In the text, ‘aberrant’ refers to the dysregulation of the expression levels of NOTCH2 and NME1. 

Now we have amended this sentence in the new revision as below.

“More interestingly, NOTCH2 and NME1 could be potential targets for m6A regulation of AD.”

About the Lines 33-34–why do you use ‘was’ and not ‘is’?

According to the reviewer’s good suggestion, we have revised the sentence using ‘is’.

And we also revised the corresponding tense thoroughly in the revised manuscript. In general, the regular contents are applied in present tense, and the methods and results are in past tense consistently in the new revision.

About Figure 1–reformulate figure legend. ‘A’ is the histogram, ‘B’ is heat map, in ‘B, you compare only 14 genes, not 25.

According to the reviewer’s good suggestion and guidance, we now have reformulated the figure legend as below for you check. And we also revised them in the new revision.

Figure 1. Expressions of m6A methylation regulators between normal controls and AD patients. (A) Differential expression boxplot of the 25 identified m6A regulators. (B) Expression heat map of the 14 differential m6A regulators. The t-test was used to analyze the difference in the expression levels of m6A regulators. Compared with the controls, *p < 0.05, ** p < 0.01, and *** p < 0.001.

About Line 95 – what does ‘cases’ mean? Cases of what?

In the text, ‘cases’ refer to the AD patients, and accordingly we now have already replaced ‘cases’ with ‘AD patients’ in the revised version. Thank you very much.

About Line 153 – not ‘blue’, but ‘green’.

We have already replaced ‘blue’ with ‘green’ in the new revised version accordingly.

About From line 178 and figure 6, the text contains wrong references to the figures, e.g. in line 179 there is ‘Figure 7A’, but it should be ‘6A’. The numbering in text does not correspond with numbering of figures 6, 7, 8.

We have checked the text and the figures carefully again and revised the reference numbers of the figures in the revision. Now in the new revised manuscript, in line 179, it was the Figure 6A and we also make the numbering correspond with the right number of Figure 6, 7 and 8 separately. Thank you very much again.

About Line 180 – What does it mean ‘key’? How were these 19 genes chosen?

We thank the reviewer very much for the good issues and kind help.

About the ‘key’, it means ‘the most important’. The key AD genes are closely related to AD and may serve as its potential biomarkers. The abnormal expression levels of key genes would affect the development and progress of the AD.

In order to choose the key 19 genes, we have done a series of analysis and screening as follows for your check.

First, we collected and merged the GSE5281 and GSE48350 microarray datasets from the same platform in the GEO database for the differential expression profile analysis. We had obtained a total of 194 differentially expressed genes (DEGs) from the merging dataset under the threshold of |log2FC| >0.585 and FDR<0.05. Then, we further profiled and identified the co-expression genes related with AD, using the weighted gene co-expression network analysis (WGCNA), and combined 194 DEGs to obtain the 42 intersecting genes with differentially expressed AD hub genes. Finally, we had screened the 19 key AD genes from these 42 genes based on the best λ values, using least absolute shrinkage and selection operator (LASSO) regression analysis.

Now accordingly, we have also replenished the relative contents in the new revision. Thank you very much.

About Line 270 – What does word ‘two’ refer to? ‘Two’ what?

The word ‘two’ in the text refers to two m6A patterns (cluster A and cluster B), and now we have replaced ‘two m6A’ with ‘two m6A patterns’ in the new revised version accordingly. Thank you very much.

About Line 368 – Table 1, not Table 2.

We thank the reviewer very much for the earnest and serious suggestion. Accordingly, we have already updated the table number with Table 1 in line 368 in the revised version.

  1. In general, in figure legends are missing informations about what statistical test was/were used and information on significances.

→ We thank the reviewer for the good suggestion and strict request. Now in the new revised manuscript, we have already added the statistical test and the significances information to all the legends. Meanwhile, we also supplemented the relatively statistical test in the Materials and Methods. Please check them in the new revision. Thank you very much.

Please check the supplemented contents of the figure legends in the new revision.

Now we also show the relatively added contents in the Materials and Methods here in red color for your check conveniently.

The t-text to compare the expression levels of m6A regulators between different two groups. The Wilcoxon rank sum test was performed to the Braak stage differences. The correlations within the continuous variables were analyzed by the Pearson’s test. All calculations in the study were dependent on R language (version 4.4.1) and Microsoft Excel 2016. The statistical significance was defined as a two-sided p<0.05.

  1. In general, when you introduce acronym for the first time you need to include full wording of the term. For example in line 168, GS and MM.

→ We thank the reviewer very much. Now in the revised version, we have checked every acronym and supplied the complete wording of each term for the first time in the new revised manuscript. Thank you again. At the meanwhile, in line 168, we also reinterpreted GS and MM as: gene significance and module membership separately in the revision.

Reviewer 2 Report

Dear Authors,

I read the manuscript entitled "Novel roles of RNA m6A methylation regulators in the occurrence of Alzheimer's disease and the subtype classification". This original study focuses on analyzing the combined role of 25 common regulators of RNA methylation, taking into account that N6-methyladenosine (m6A) is the most abundant form that contributes to RNA modification with implications pathological in Alzheimer's Dementia (AD).

The work is valuable because it analyzed gene expression data on brain tissue from 250 normal controls and 176 AD patients using bioinformatic tests.

The paper complies with the requirements of the journal, being based on a thorough and rigorous analysis, very well structured. Each section was presented in a detailed way, the results obtained were highlighted by means of 8 figures, giving the possibility to the reader to make the reading easier. The bibliographic references used for documentation are in accordance with the chosen topic.

I have a few minor observations though:

1. In the Discussion section, line 255, you mentioned that "It might suggest that m6A modification was generally upregulated in AD brain". Could the authors provide some insight into this phenomenon?

2. In section Material and Methods, you presented Table no. 2. I presume that there would also be a Table with no. 1 that does not appear in the manuscript. Or maybe it was just a table numbering error?

3. Indeed, the requirements of the journal do not impose a conclusion section. But for a better systematization of the information and increasing the scientific value of the manuscript, it would be indicated and I suggest you introduce a final section of conclusions.

4. For the 8 figures, I recommend using a larger font size to increase readability.

5. The English language could be improved.

Author Response

Reviewer 2:

I read the manuscript entitled "Novel roles of RNA m6A methylation regulators in the occurrence of Alzheimer's disease and the subtype classification". This original study focuses on analyzing the combined role of 25 common regulators of RNA methylation, taking into account that N6-methyladenosine (m6A) is the most abundant form that contributes to RNA modification with implications pathological in Alzheimer's Dementia (AD).

The work is valuable because it analyzed gene expression data on brain tissue from 250 normal controls and 176 AD patients using bioinformatic tests.

The paper complies with the requirements of the journal, being based on a thorough and rigorous analysis, very well structured. Each section was presented in a detailed way, the results obtained were highlighted by means of 8 figures, giving the possibility to the reader to make the reading easier. The bibliographic references used for documentation are in accordance with the chosen topic. I have a few minor observations though:

  1. In the Discussion section, line 255, you mentioned that "It might suggest that m6A modification was generally upregulated in AD brain". Could the authors provide some insight into this phenomenon?

→ We thank the reviewer very much for the good and constructive suggestion. Now we have already rewritten the sentence more exactly, and also supplied the application prospect and the theory meaning of this phenomenon in the discussion of the new revision. We also show them here for you check in red color.

Most of the m6A methylation binding proteins, including YTHDC1, YTHDF1, FMR1, IGF2BP2 and HNRNPA2B1, were elevated in AD brain tissues, suggesting that m6A modifications of specific target genes were generally increased in AD brain tissues. Differential levels of these m6A regulators would bring the abnormal epi-transcriptome microenvironment, also affect the expression of target genes, and ultimately contribute to the onset and progress of AD. The m6A modifications played key roles in the synaptic function of brain neurons, as the main mechanisms of AD development. However, m6A and the intrinsic regulatory mechanisms remain blank in AD. Our results have provided the theoretical basis and possible research di-rection of m6A to the etiology, clinical application, and potential molecular therapy for AD.

  1. In section Material and Methods, you presented Table no. 2. I presume that there would also be a Table with no. 1 that does not appear in the manuscript. Or maybe it was just a table numbering error?

→ We thank the reviewer very much for the good suggestion and warm guidance. We fully apologize for the numbering error in the table. Yes, it should be Table 1. Now accordingly, we have corrected all the numbering and made them in the right positions in the new revised manuscript. Thank you again.

  1. Indeed, the requirements of the journal do not impose a conclusion section. But for a better systematization of the information and increasing the scientific value of the manuscript, it would be indicated and I suggest you introduce a final section of conclusions.

→ We thank the reviewer very much for the constructive suggestion. Accordingly, for a better systematization of the information and increasing the scientific value, we have already supplemented a final section of conclusions in the new revision. We also show them here for your check as follows in red color.

In conclusion, the m6A methylation regulators have played crucial roles in the occurrence, subtype classification, and immune infiltration of AD. There were 14 m6A regulators associated with the development of AD, and then AD patients could be well classified into two categories, one of which was closely associated with Th1 dominated immunity with a higher Braak staging. Meanwhile, HNRNPA2B1, YTHDF1, IGF2BP2A, and FTO were closely associated with AD immune cell components. The m6A regulators and AD key genes have close connections, with NOTCH2 and NME1 as the potentially specific modification targets, providing the new clues for the precise prevention and intervention of AD. It is very necessary to explore more detailed mechanisms of m6A methylation in AD occurrence and immunity changes in future researches.

  1. For the 8 figures, I recommend using a larger font size to increase readability.

→ We thank the reviewer very much for the good guidance and help. Now we have already offered revised figures with a larger font size to increase readability accordingly in the new revision.

  1. The English language could be improved.

→ We are grateful to the reviewer for the good suggestion and kind patience. Now we have checked the manuscript thoroughly and carefully, and revised them sentence by sentence in the new revision. Thank you very much indeed!
